# SynGR: Unleashing the Potential of Cross-Modal Synergy for Generative Recommendation

**Wei Chen** [1*]   **Xingyu Guo** [1*]   **Shuang Li** [1†]   **Fuwei Zhang** [1]   **Meng Yuan** [1]   **Jing Fan** [1]
**Zhao Zhang** [2]   **Deqing Wang** [2]   **Fuzhen Zhuang** [1†]

## Abstract

Generative Recommendation (GR) has emerged as a promising paradigm by formulating item recommendation as a sequence-to-sequence generation task over item identifiers. Recent studies have incorporated multimodal signals to provide richer token-level evidence for generation. However, existing approaches largely rely on alignment-centric fusion and underexplore synergistic information across modalities. In practice, synergistic information plays a critical role in capturing emergent item properties that cannot be inferred from any single modality alone. Such properties encode intrinsic item semantics and guide user preferences, enabling models to move beyond surface-level feature matching. To address this limitation, we propose **SynGR**, a synergistic generative recommendation framework that explicitly encourages the exploitation of cross-modal dependencies during generation. By constraining overreliance on dominant modalities, SynGR enables the model to capture emergent item semantics beyond shared or modality-specific signals. Extensive experiments across three benchmark datasets demonstrate that SynGR achieves superior performance. Code is available at: SynGR.

## 1. Introduction

The evolution of recommender systems has recently shifted from traditional discriminative ranking to Generative Recommendation (GR) (Ko et al., 2022; Ji et al., 2024; Zhao et al., 2025; Chen et al., 2024; 2025). Unlike conventional

*Equal contribution  [1]School of Artificial Intelligence, Beihang University, Beijing, China  [2]School of Computer Science and Engineering, Beihang University, Beijing, China. Correspondence to: Fuzhen Zhuang <zhuangfuzhen@buaa.edu.cn>, Shuang Li <shuangliai@buaa.edu.cn>.

*Proceedings of the 43rd International Conference on Machine Learning*, Seoul, South Korea. PMLR 306, 2026. Copyright 2026 by the author(s).

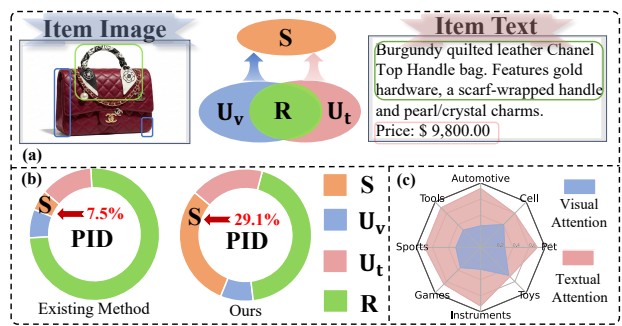

*Figure 1.* **(a)** Illustration of cross-modal information decomposition, where $\mathbf{S}, \mathbf{R}, \mathbf{U_t}$ and $\mathbf{U_v}$ denote **synergistic**, **redundant**, and modality-specific **unique** information, respectively. **(b)** Comparison of synergistic components estimated using a normalized PID-inspired performance decomposition (Kolchinsky, 2022). **(c)** The distribution of visual and textual attention across datasets.

methods that map users and items to latent embeddings for heuristic scoring, GR formulates user behavior modeling as a sequence-to-sequence generation task (Li et al., 2024a; 2025a). By representing items with discrete semantic identifiers, such as quantized tokens or hierarchical IDs (Qu et al., 2025), these frameworks utilize the sequential reasoning power of Pre-trained Language Models (PLMs) to capture complex and evolving user preferences (Hou et al., 2024; He et al., 2025; Yuan et al., 2025; Huang et al., 2025).

Recent studies (Liu et al., 2024a; Wang et al., 2025) in GR have begun to incorporate multimodal signals, such as visual and textual information, to provide richer item evidence and improve the generation of discrete semantic identifiers. However, the practical gains of these approaches remain limited. Most existing methods (Zhu et al., 2025; Zhai et al., 2025; Zhang et al., 2026a) rely on alignment-centric fusion strategies that enforce cross-modal consistency through shared representations. While such designs are effective for basic semantic matching, they primarily encourage modality agreement rather than modality interaction, and consequently struggle to capture intricate synergistic information that arises only from the joint interplay across modalities.

Figure 1(a) illustrates the critical role of synergistic information **S** in capturing emergent item properties that are essen-

tial for accurate recommendation. Taking a luxury handbag as an example, the visual modality $\mathbf{U_v}$ captures aesthetic nuances such as quilted leather textures and fine-grained embellishments, while the textual modality $\mathbf{U_t}$ provides explicit signals like brand identity and pricing. Individually, neither modality is sufficient to characterize the item's full semantic profile. Instead, their synergy $\mathbf{S}$ facilitates the emergence of higher-level attributes, such as brand prestige and luxury positioning, which are unattainable from any single modality alone. These emergent properties constitute the intrinsic semantics that shape user preferences, enabling the model to move beyond surface-level feature matching toward a deeper understanding of user intent. Despite its importance, the synergistic component $\mathbf{S}$ remains insufficiently explored in existing generative paradigms, which often emphasize redundant or modality-unique signals. As an illustrative case, our empirical analysis in Figure 1(b) reveals that even a representative state-of-the-art method, MACRec (Zhang et al., 2026a), captures a mere 7.5% of its learned representation as synergistic information on the Arts dataset[1]. This observation suggests a systemic challenge in current multimodal GR: the inherent difficulty of effectively **unleashing the latent potential of cross-modal synergy**.

This limitation stems from inherent disparities between modalities under the generative training objective. Textual inputs are discrete, sequential, and semantically compact, with individual tokens often encoding highly discriminative signals such as brand, category, or style, whereas visual embeddings are continuous, high-dimensional, and noisy, requiring significant aggregation to manifest coherent semantics (Parcalabescu & Frank, 2023). When such heterogeneous modalities are mapped into codebooks of comparable capacity, the fixed codebook budget represents fundamentally unequal semantic depth, leading to a mismatch in effective information density. In recommendation scenarios, this imbalance allows certain modalities to provide a more direct and reliable mapping to the target identifier, forming a path of least resistance for minimizing the generation loss (Lin et al., 2024). Consistent with this behavior, Figure 1(c) reveals that across multiple datasets, the current GR model assigns disproportionately higher attention weights to the textual modality, biasing learning toward text-dominant representations during next-token prediction. Consequently, complementary evidence from secondary modalities is progressively deemphasized. Since synergistic information requires high-order interactions, this unimodal convergence suppresses the emergence of complex multimodal patterns.

Motivated by these observations, we argue that bridging the synergy gap requires explicit intervention to prevent generative models from collapsing into unimodal shortcuts. To this end, we propose **SynGR**, a synergistic generative

recommendation framework that actively disrupts the path of least resistance during training. The key idea is to constrain the model's reliance on dominant modalities, thereby compelling it to leverage higher-order cross-modal dependencies to satisfy the generative objective. SynGR achieves this through a saliency-aware masking mechanism that adaptively attenuates dominant signals and encourages the exploration of latent synergistic information across modalities. To further stabilize and refine these emergent semantics, the framework incorporates a synergy-oriented contrastive objective that explicitly promotes cross-modal interaction. Our main contributions are summarized as follows:

(i) **Insight.** We identify a synergy gap in generative recommendation, showing that information density mismatch across modalities induces unimodal shortcuts and suppresses higher-order cross-modal semantics. (ii) **Method.** We propose SynGR, a synergistic generative recommendation framework that explicitly disrupts unimodal shortcuts and compels the model to leverage cross-modal dependencies during generation. (iii) **Evaluation.** We demonstrate through extensive experiments that SynGR consistently enhances synergistic learning and improves recommendation performance across three real-world datasets, achieving average gains of around 10% over strong baselines.

## 2. Related Work

**Sequential Recommendation.** Sequential recommendation (Boka et al., 2024) aims to capture the evolution of user interests from historical interaction logs. Early research predominantly focused on temporal dependencies using Recurrent Neural Networks (RNNs) (Medsker et al., 2001), with pioneering models such as GRU4Rec (Hidasi et al., 2016) and NARM (Li et al., 2017) establishing the foundation for session-based modeling. To address the limitations of RNNs in capturing long-range dependencies, SASRec (Kang & McAuley, 2018) introduced self-attention mechanisms, while BERT4Rec (Sun et al., 2019) employed a bidirectional Cloze objective to enhance sequence representations. This field has recently shifted toward a unified paradigm where recommendation is treated as a language modeling task. For instance, P5 (Geng et al., 2022; Tao et al., 2025) leveraged PLMs to convert diverse recommendation tasks into a standardized text-to-text format. To further alleviate data sparsity, researchers extended this paradigm to the multimodal domain. Notably, VIP5 (Geng et al., 2023) served as a multimodal foundation model by integrating visual and textual signals into the generative framework.

**Generative Recommendation.** The advent of Large Language Models (LLMs) (Naveed et al., 2025; Zhang et al., 2025b;a; Li et al., 2024b; Yang et al., 2026a; Li et al., 2026; Cao et al., 2026; Liang et al., 2026; Yang et al., 2026b) has pivoted item retrieval toward Generative Recommendation

---

[1]The calculation details are shown in Appendix C.

(GR), redefining it as a conditional next-token prediction task. TIGER (Rajput et al., 2023) pioneered this paradigm by quantizing items into hierarchical semantic IDs via RQ-VAE. Subsequent optimizations include LC-Rec (Zheng et al., 2024), which harnessed LLM reasoning for fine-tuning, and LETTER (Wang et al., 2024), which integrated collaborative signals to enhance codebook distinctiveness. The frontier has recently shifted toward multimodal GR. MMGRec (Liu et al., 2024a) and MQL4GRec (Zhai et al., 2025; Li et al., 2025b) explored multimodal blending and cross-domain knowledge transfer through quantized languages. More recently, MACRec (Zhang et al., 2026a) introduced a multi-aspect quantization framework to mitigate semantic loss and capture complementary information via explicit alignment. Despite these advances, existing methods primarily focus on alignment or fusion while overlooking information synergy between modalities (Liu et al., 2024b). Excessive redundancy can overshadow cross-modal synergy, which refers to emergent, non-redundant information arising from multimodal interactions. Our SynGR addresses this issue by explicitly promoting synergistic information while suppressing redundant signals.

## 3. Theoretical Analysis

In this section, we provide a formal information-theoretic analysis of multimodal GR. Specifically, we analyze why alignment-based methods fall short in capturing cross-modal interactions and formalize how synergistic information can be effectively exploited for discrete token generation.

### 3.1. Problem Setup

Let $\mathcal{H} = \{\mathbf{x}_1, \mathbf{x}_2, \ldots, \mathbf{x}_L\}$ denote a user's historical interaction sequence of length $L$. Each item is associated with visual tokens $\mathbf{x}_{v,i}$ and textual tokens $\mathbf{x}_{t,i}$. In GR, the multimodal histories $\mathbf{X_v} = \{\mathbf{x}_{v,1}, \ldots, \mathbf{x}_{v,L}\}$ and $\mathbf{X_t} = \{\mathbf{x}_{t,1}, \ldots, \mathbf{x}_{t,L}\}$ are treated as model inputs. The objective is to learn a generative model that predicts the discrete semantic identifier of the next item $\mathbf{Y}$ by modeling the conditional distribution $P(\mathbf{Y} \mid \mathbf{X_v}, \mathbf{X_t})$. From the information-theoretic perspective (Lin et al., 1998), this corresponds to maximizing the task-relevant information that the joint multimodal history provides about the target $\mathbf{Y}$, which is quantified by the mutual information $I(\mathbf{X_v}, \mathbf{X_t}; \mathbf{Y})$.

### 3.2. Partial Information Decomposition Framework

To analyze how different modalities in the user history contribute to the prediction of $\mathbf{Y}$, we use the Partial Information Decomposition (PID) framework (Kolchinsky, 2022; Dufumier et al., 2025) to decompose the multivariate mutual information $I(\mathbf{X_v}, \mathbf{X_t}; \mathbf{Y})$ into three distinct types:

(1) **Uniqueness** ($\mathbf{U_v}, \mathbf{U_t}$): This represents information pro-

vided exclusively by one modality stream across the history. For instance, $\mathbf{U_t}$ captures purely textual patterns in user behavior that help predict the next item without needing visual cues. (2) **Redundancy** ($\mathbf{R}$): This refers to the task-relevant information shared between the visual and textual histories, such as the consistent category or brand preferences reflected in both images and descriptions. (3) **Synergy** ($\mathbf{S}$): This is the emergent information that appears only when $\mathbf{X_v}$ and $\mathbf{X_t}$ are integrated. Synergistic semantics (Zhang et al., 2026b) represent complex user preferences, such as a specific aesthetic style combined with a functional price point, which cannot be decoded from either modality sequence in isolation. The total information provided by the multimodal sequence about the target $\mathbf{Y}$ is expressed as:

$$I(\mathbf{X_v}, \mathbf{X_t}; \mathbf{Y}) = \mathbf{R} + \mathbf{S} + \mathbf{U_v} + \mathbf{U_t}. \tag{1}$$

Existing GR models (Zhai et al., 2025; Zhang et al., 2026a) typically optimize objectives that emphasize either individual modality streams or their shared semantic components. Specifically, same-modality generation aim to maximize the mutual information between each modality and the target:

$$I(\mathbf{X_v}; \mathbf{Y}) = \mathbf{R} + \mathbf{U_v}, \quad I(\mathbf{X_t}; \mathbf{Y}) = \mathbf{R} + \mathbf{U_t}, \tag{2}$$

where $\mathbf{R}$ denotes redundant information shared across modalities, and $\mathbf{U_v}$ and $\mathbf{U_t}$ represent modality-specific unique information. In parallel, cross-modality alignment and generation strategies seek to increase consistency between modalities by maximizing

$$I(\mathbf{X_v}; \mathbf{X_t}) = \mathbf{R}, \tag{3}$$

thereby primarily reinforcing redundant semantics. However, under the PID framework, the total information captured by these objectives remains smaller than the joint mutual information available in the multimodal history:

$$I(\mathbf{X_v}; \mathbf{Y}) + I(\mathbf{X_t}; \mathbf{Y}) - I(\mathbf{X_v}; \mathbf{X_t}) \; < \; I(\mathbf{X_v}, \mathbf{X_t}; \mathbf{Y}). \tag{4}$$

This inequality reveals a fundamental limitation of existing paradigms: while redundancy and uniqueness information are effectively captured, the synergistic component $\mathbf{S}$ is entirely absent. Thus, existing objectives do not explicitly target synergistic semantics and may under-utilize information that arises only through deep cross-modal integration.

### 3.3. Isolate Synergistic Information

To bridge the gap identified by PID, we propose a theoretical framework to isolate the synergistic component $\mathbf{S}$ by constructing a synergy-preserving representation. We begin by defining a transformation family $\Phi$ that acts on the multimodal input $(\mathbf{X_v}, \mathbf{X_t})$. The goal is to derive a synergistic view $\widehat{\mathbf{X}}$ that discards modality-specific shortcuts while preserving joint predictive power (Wen et al., 2025).

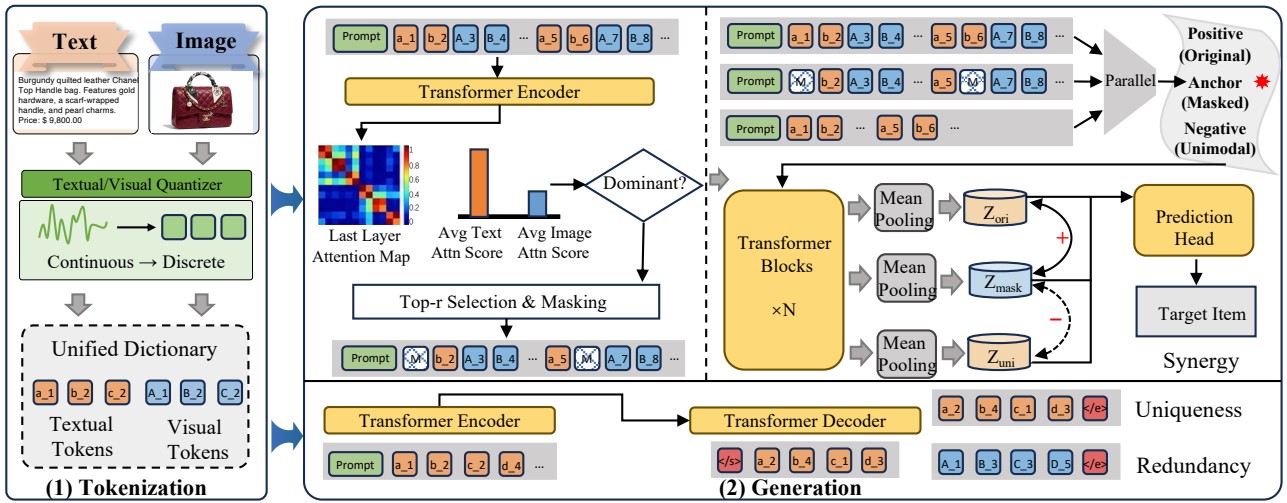

*Figure 2.* Overview of the proposed SynGR framework. (1) In the tokenization phase, continuous textual and visual features are discretized into a unified dictionary through respective quantizers. (2) The generation phase begins with saliency diagnosis, where a transformer encoder extracts attention maps to identify dominant modalities for adaptive top-r masking. Subsequently, the original, masked, and unimodal sequences are processed in parallel through transformer blocks to derive the embeddings $\mathbf{Z}_{\text{ori}}$, $\mathbf{Z}_{\text{mask}}$, and $\mathbf{Z}_{\text{uni}}$ for synergistic contrastive learning. Finally, these representations are optimized via a transformer decoder to capture synergistic information.

**Definition 1 (Synergy-oriented Transformation).** A transformation $\phi$ is said to be synergy-oriented if the resulting view $\widehat{\mathbf{X}} = \phi(\mathbf{X_v}, \mathbf{X_t})$ satisfies the following two properties:

**(1) Joint Sufficiency:** $\mathrm{I}(\widehat{\mathbf{X}}; \mathbf{Y}) \approx \mathrm{I}(\mathbf{X_v}, \mathbf{X_t}; \mathbf{Y})$, ensuring that the transformation does not substantially discard task-relevant predictive information for the target $\mathbf{Y}$.

**(2) Unimodal Obscuration:** The transformation attenuates reliance on individual streams by enforcing:

$$\mathrm{I}(\widehat{\mathbf{X}}; \mathbf{X_v}) \leq \epsilon \cdot \mathrm{I}(\mathbf{X_v}; \mathbf{Y}), \quad \mathrm{I}(\widehat{\mathbf{X}}; \mathbf{X_t}) \leq \epsilon \cdot \mathrm{I}(\mathbf{X_t}; \mathbf{Y}), \quad (5)$$

where $\epsilon > 0$ is a sufficiently small threshold. This condition bounds the information leakage from any single modality relative to task-relevant signals.

In practice, these conditions can be approximated through full-history encoding and shortcut-suppression mechanisms such as modality masking and contrastive regularization. Under these principles, the model is encouraged to reconstruct $\mathbf{Y}$ by leveraging latent cross-modal interactions rather than unimodal cues. Accordingly, we consider the following Markov chain (Brooks, 1998) to characterize the information flow in the encoding process:

$$(\mathbf{X_v}, \mathbf{X_t}) \xrightarrow{\phi} \widehat{\mathbf{X}} \xrightarrow{\text{Transformer}} \mathbf{Z}_{\text{syn}} \xrightarrow{\text{Predictor}} \mathbf{Y}. \quad (6)$$

**Lemma 1.** *Let $\mathbf{Z}_{syn}$ be a representation learned from the synergistic view $\widehat{\mathbf{X}}$. If $\mathbf{Z}_{syn}$ is optimized to remain highly predictive of $\mathbf{Y}$ while substantially suppressing information recoverable from individual modalities, then $\mathbf{Z}_{syn}$ is dominated by the synergistic component $\mathbf{S}$.*

**Proof.** The complete proof is provided in Appendix A. □

**How Could Theory Further Drive Practice?** These analyses motivate a synergy-oriented optimization objective:

$$\max \mathrm{I}(\mathbf{Z}_{\text{syn}}; \mathbf{Y}) \quad \text{s.t.} \quad \min \sum_{m \in \{\mathbf{v}, \mathbf{t}\}} \mathrm{I}(\mathbf{Z}_{\text{syn}}; \mathbf{Z}_m), \quad (7)$$

where $\mathrm{I}(\mathbf{Z}_{\text{syn}}; \mathbf{Y})$ represents the predictive power derived from synergistic signals, and $\mathbf{Z}_m$ denotes the unimodal representations for $m \in \{\mathbf{v}, \mathbf{t}\}$. By maximizing mutual information with the target $\mathbf{Y}$ while suppressing information from individual modalities, the objective encourages the model to capture the missing synergistic signals.

## 4. Framework Overview

Motivated by the theoretical analysis in Section 3.3, we propose **SynGR**, illustrated in Figure 2, a framework designed to explicitly capture cross-modal semantics.

### 4.1. Multimodal Tokenization via RQ-VAE

To bridge the gap between continuous multimodal representations and the discrete input space required by generative models, we adopt a Residual-Quantized Variational AutoEncoder (RQ-VAE) (Lee et al., 2022) to discretize item content. For each item $i$, continuous embeddings from different modalities are first extracted using frozen encoders, such as LLaMA for text (Touvron et al., 2023) and ViT for vision (Yuan et al., 2021). These embeddings are then projected into modality-specific latent vectors $\mathbf{z}_m$, where $m \in \mathbf{t}, \mathbf{v}$.

To achieve high-fidelity discretization, each latent vector $\mathbf{z}_m$

is encoded through a $D$-level residual quantization process. Specifically, let $\mathcal{C}^d = \{\mathbf{e}_k^d\}_{k=1}^K$ denotes the codebook at quantization depth $d \in 1, \ldots, D$, where $K$ is the codebook size. At each depth, the residual is quantized by selecting the nearest codeword, and the residual is updated accordingly:

$$
\begin{aligned}
c_{m,d} &= \arg\min_k \|\mathbf{r}_{m,d-1} - \mathbf{e}_k^d\|_2^2, \\
\mathbf{r}_{m,d} &= \mathbf{r}_{m,d-1} - \mathbf{e}_{c_{m,d}}^d,
\end{aligned}
\tag{8}
$$

where $\mathbf{r}_{m,0} = \mathbf{z}_m$. This recursive formulation allows the model to progressively capture fine-grained semantic information while maintaining a compact discrete representation. To enable unified sequence modeling while preserving modality identity, we construct a shared vocabulary $\mathcal{V} = \mathcal{V}_\mathbf{t} \cup \mathcal{V}_\mathbf{v}$ with modality-specific and depth-aware prefixes following (Zhai et al., 2025; Guo et al., 2025). Concretely, textual tokens are assigned lowercase prefixes, whereas visual tokens use uppercase counterparts, ensuring that the generative backbone can distinguish modalities within a single token stream. Assuming a quantization depth of $D = 3$, the resulting discrete identifier for item $i$ is given by $\mathbf{x}_i = [a_1, b_2, c_3, A_1, B_2, C_3]$. This unified sequence allows model to process multimodal content as a single, coherent stream for modeling cross-modal interactions. It should be noted that, as our primary objective is to enhance cross-modal synergy during the generative stage, we omit the training specifics of the tokenization process.

## 4.2. Unleashing Synergistic Information

Building upon the generative identifiers, this section describes how the proposed synergy isolation framework is instantiated in practice. To realize the synergy-preserving transformation $\phi$, we construct synergy-preserving views through a saliency-aware masking mechanism that selectively suppresses shortcut-dominant features. Furthermore, we introduce a synergistic contrastive learning objective that explicitly encourages the preservation of the synergistic component $\mathbf{S}$ within the generative latent space.

### 4.2.1. SALIENCY-AWARE MASKING MECHANISM

To satisfy the Unimodal Obscuration property of $\phi$ defined in Definition 1, an effective strategy is to selectively mask tokens belonging to the dominant modality. By attenuating these highly predictive yet redundant unimodal shortcuts, the model is prevented from relying solely on individual stream cues and is instead compelled to reconstruct item semantics through cross-modal interactions, thereby facilitating the isolation of the synergistic component $\mathbf{S}$.

Formally, let $\mathcal{H} = \{\mathbf{x}_1, \ldots, \mathbf{x}_N\}$ be the multimodal input sequence, where $N$ denotes total token count. To diagnose modality-specific shortcuts, we extract the self-attention weights from the final encoder layer, denoted as $\{\mathbf{A}^{(m)} \in$

$\mathbb{R}^{N \times N}\}_{m=1}^M$ for $M$ attention heads. We then compute a global saliency score (Abnar & Zuidema, 2020) $\boldsymbol{\ell}_i$ for each token by aggregating its total received attention mass:

$$
\boldsymbol{\ell}_i = \frac{1}{M \cdot N} \sum_{m=1}^M \sum_{j=1}^N \mathbf{A}_{j,i}^{(m)}.
\tag{9}
$$

The denominator $M \cdot N$ serves as a normalization factor across all heads and query positions, providing a robust, model-internal estimate of token $i$'s relative influence on the final contextual representation.

Building upon these scores, we define the modality-level saliency densities for text ($\bar{\boldsymbol{\ell}}_\mathbf{t}$) and vision ($\bar{\boldsymbol{\ell}}_\mathbf{v}$) as follows:

$$
\bar{\boldsymbol{\ell}}_\mathbf{t} = \frac{1}{|\mathbf{X}_\mathbf{t}|} \sum_{i \in \mathbf{X}_\mathbf{t}} \boldsymbol{\ell}_i, \quad \bar{\boldsymbol{\ell}}_\mathbf{v} = \frac{1}{|\mathbf{X}_\mathbf{v}|} \sum_{i \in \mathbf{X}_\mathbf{v}} \boldsymbol{\ell}_i.
\tag{10}
$$

The sets $\mathbf{X}_\mathbf{t}$ and $\mathbf{X}_\mathbf{v}$ contain the indices of tokens belonging to the textual and visual modalities, respectively. The modality exhibiting the higher average saliency is identified as the dominant modality, $\mathcal{M}_{\text{dom}} = \arg\max\{\bar{\boldsymbol{\ell}}_\mathbf{t}, \bar{\boldsymbol{\ell}}_\mathbf{v}\}$. Finally, we mask the top-r salient tokens within the identified dominant modality $\mathcal{M}_{\text{dom}}$:

$$
\tilde{\mathbf{x}}_i = \begin{cases} \texttt{[MASK]}, & \text{if } i \in \text{top-r}(\mathcal{M}_{\text{dom}}), \\ \mathbf{x}_i, & \text{otherwise.} \end{cases}
\tag{11}
$$

This structured intervention forces the generator to bridge the information gap through synergistic reasoning rather than relying on isolated unimodal cues.

### 4.2.2. SYNERGISTIC-AWARE CONTRASTIVE LEARNING

To implement Eq. (7), we propose a synergistic contrastive learning scheme that explicitly promotes cross-modal interaction while suppressing unimodal shortcuts. Specifically, for each interaction history, we construct a triplet of semantic views to isolate the synergistic component $\mathbf{S}$:

- **Anchor View** ($\mathcal{H}_{\text{mask}}$): The masking sequence where dominant modality shortcuts are selectively masked.
- **Positive View** ($\mathcal{H}_{\text{ori}}$): The original multimodal sequence containing holistic item semantics.
- **Negative Views** ($\mathcal{H}_{\text{uni}}$): A unimodal sequence set.

To transform the token-wise hidden states into a global sequence-level representation, we apply a mask-aware pooling operation to the decoder's final output (Reimers & Gurevych, 2019). Let $\mathbf{h}_t^{\text{dec}} \in \mathbb{R}^d$ denote the hidden vector at position $t$ within the sequence of length $N$. The pooled representation $\mathbf{Z}$ is formulated as:

$$
\mathbf{Z} = \frac{\sum_{t=1}^N \mathbb{I}(y_t \neq \texttt{[PAD]}) \cdot \mathbf{h}_t^{\text{dec}}}{\sum_{t=1}^N \mathbb{I}(y_t \neq \texttt{[PAD]}) + \xi},
\tag{12}
$$

*Table 1.* Overall performance comparison across benchmark datasets. Best and second-best results are indicated in **bold** and underlined fonts respectively. Red font denotes the relative improvement of SynGR over the strongest baseline.

| Model / Datasets | Arts | | | | | Games | | | | | Instruments | | | | |
|---|---|---|---|---|---|---|---|---|---|---|---|---|---|---|---|
| | HR@1 | HR@5 | HR@10 | N@5 | N@10 | HR@1 | HR@5 | HR@10 | N@5 | N@10 | HR@1 | HR@5 | HR@10 | N@5 | N@10 |
| GRU4Rec (ICLR'16) | 0.0365 | 0.0817 | 0.1088 | 0.0602 | 0.0690 | 0.0140 | 0.0544 | 0.0895 | 0.0341 | 0.0453 | 0.0566 | 0.0975 | 0.1207 | 0.0783 | 0.0857 |
| SASRec (ICDM'18) | 0.0212 | 0.0951 | 0.1250 | 0.0610 | 0.0706 | 0.0069 | 0.0587 | 0.0985 | 0.0333 | 0.0461 | 0.0318 | 0.0946 | 0.1233 | 0.0654 | 0.0746 |
| BERT4Rec (CIKM'19) | 0.0289 | 0.0697 | 0.0922 | 0.0502 | 0.0575 | 0.0115 | 0.0426 | 0.0725 | 0.0270 | 0.0366 | 0.0450 | 0.0856 | 0.1081 | 0.0667 | 0.0739 |
| FDSA (IJCAI'19) | 0.0380 | 0.0832 | 0.1190 | 0.0583 | 0.0695 | 0.0163 | 0.0614 | 0.0988 | 0.0389 | 0.0509 | 0.0530 | 0.0987 | 0.1249 | 0.0775 | 0.0859 |
| S³-Rec (CIKM'20) | 0.0172 | 0.0739 | 0.1030 | 0.0511 | 0.0630 | 0.0136 | 0.0527 | 0.0903 | 0.0351 | 0.0468 | 0.0339 | 0.0937 | 0.1123 | 0.0693 | 0.0743 |
| P5-CID (RecSys'22) | 0.0421 | 0.0713 | 0.0994 | 0.0607 | 0.0662 | 0.0169 | 0.0532 | 0.0824 | 0.0331 | 0.0454 | 0.0512 | 0.0839 | 0.1119 | 0.0678 | 0.0704 |
| VQ-Rec (WWW'23) | 0.0408 | 0.1038 | 0.1386 | 0.0732 | 0.0844 | 0.0075 | 0.0408 | 0.0679 | 0.0242 | 0.0329 | 0.0502 | 0.1062 | 0.1357 | 0.0796 | 0.0891 |
| MISSRec (MM'23) | 0.0479 | 0.1021 | 0.1321 | 0.0699 | 0.0815 | 0.0201 | 0.0674 | 0.1048 | 0.0385 | 0.0499 | 0.0723 | 0.1089 | 0.1361 | 0.0797 | 0.0880 |
| VIP5 (EMNLP'23) | 0.0474 | 0.0704 | 0.0859 | 0.0586 | 0.0635 | 0.0173 | 0.0480 | 0.0758 | 0.0328 | 0.0418 | 0.0737 | 0.0892 | 0.1071 | 0.0815 | 0.0872 |
| TIGER (NeurIPS'23) | 0.0532 | 0.0894 | 0.1167 | 0.0718 | 0.0806 | 0.0166 | 0.0523 | 0.0857 | 0.0345 | 0.0453 | 0.0754 | 0.1007 | 0.1221 | 0.0882 | 0.0950 |
| MQL4GRec (ICLR'25) | 0.0672 | 0.1037 | 0.1327 | 0.0857 | 0.0950 | 0.0203 | 0.0637 | 0.1033 | 0.0421 | 0.0548 | 0.0833 | 0.1115 | 0.1375 | 0.0977 | 0.1060 |
| MACRec (AAAI'26) | 0.0685 | 0.1046 | 0.1329 | 0.0868 | 0.0953 | 0.0208 | 0.0671 | 0.1078 | 0.0435 | 0.0565 | 0.0819 | 0.1110 | 0.1363 | 0.0965 | 0.1046 |
| **SynGR(Ours)** | **0.0713** | **0.1145** | **0.1449** | **0.0919** | **0.1010** | **0.0245** | **0.0702** | **0.1092** | **0.0471** | **0.0596** | **0.0850** | **0.1321** | **0.1768** | **0.1045** | **0.1189** |
| Improv. (%) | +4.09% | +9.46% | +4.55% | +5.88% | +5.98% | +17.79% | +4.15% | +1.30% | +8.28% | +5.49% | +2.04% | +18.48% | +28.58% | +6.96% | +12.17% |

where $t \in [1, N]$ denotes the token position, $y_t$ is the token identifier at position $t$, and [PAD] is the padding token used for batch alignment. The indicator function $\mathbb{I}(\cdot)$ excludes padding positions from the aggregation, and $\xi$ is a small constant (set to $1e-7$) introduced for numerical stability.

By applying this pooling to the hidden states of three input sequences respectively, we obtain the holistic representation $\mathbf{Z}_{\mathrm{ori}}$, the masking anchor $\mathbf{Z}_{\mathrm{mask}}$, and the unimodal shortcut representations $\mathbf{Z}_{\mathrm{uni}}$, respectively. The learning objective is operationalized via a tripartite contrastive loss (Wang & Liu, 2021) that regularizes the latent space, incentivizing the decoder to recover holistic semantics from the anchor while distancing it from unimodal shortcut distributions, that is:

$$\mathcal{L}_{\mathrm{Syn}} = -\log \frac{e^{\mathrm{sim}(\mathbf{Z}_{\mathrm{mask}}, \mathbf{Z}_{ori})/\tau}}{e^{\mathrm{sim}(\mathbf{Z}_{\mathrm{mask}}, \mathbf{Z}_{\mathrm{ori}})/\tau} + e^{\mathrm{sim}(\mathbf{Z}_{\mathrm{mask}}, \mathbf{Z}_{\mathrm{uni}})/\tau}}, \quad (13)$$

where $\tau$ is a temperature parameter, $\mathrm{sim}(\cdot, \cdot)$ denotes the cosine similarity in the latent space.

**Remark.** This objective $\mathcal{L}_{\mathrm{Syn}}$ is computed symmetrically for both the textual and visual sub-tasks. Specifically, for the textual sub-task, the target item is represented by sequences of textual tokens, and $\mathbf{Z}_{\mathrm{uni}}$ is derived from the unimodal textual input to serve as the shortcut representation. An identical procedure is executed for visual modality.

### 4.3. Model Optimization

SynGR is optimized via a unified objective coupling autoregressive generation with synergistic regularization. Specifically, we minimize the negative log-likelihood (NLL) (Lastras-Montaño, 2019) for next-token prediction across the tripartite views $\{\mathcal{H}_{\mathrm{ori}}, \mathcal{H}_{\mathrm{mask}}, \mathcal{H}_{\mathrm{uni}}\}$ (Sec. 4.2.2):

$$\mathcal{L}_{\mathrm{Gen}} = \sum_{\mathcal{H} \in \{\mathcal{H}_{\mathrm{ori}}, \mathcal{H}_{\mathrm{mask}}, \mathcal{H}_{\mathrm{uni}}\}} \left( -\sum_{j=1}^{|\mathbf{Y}|} \log P_\theta(y_j \mid y_{<j}, \mathcal{H}) \right), \quad (14)$$

where $\theta$ denotes model parameters, $\mathbf{Y} = \{y_1, \ldots, y_{|\mathbf{Y}|}\}$ is the target identifier sequence. The final objective integrates this generative loss with $\mathcal{L}_{\mathrm{Syn}}$ to capture synergistic $\mathbf{S}$:

$$\mathcal{L} = \mathcal{L}_{\mathrm{Gen}} + \lambda \mathcal{L}_{\mathrm{Syn}}, \quad (15)$$

where $\lambda$ is a hyperparameter. Here, $\mathcal{L}_{\mathrm{Syn}}$ serves as a regularization term that encourages the model to capture complementary cross-modal interactions beyond unimodal semantics. Meanwhile, the multi-view generative objectives inherited from prior work (Zhang et al., 2026a) help preserve redundant ($\mathbf{R}$) and modality-specific information ($\mathbf{U}$), preventing the degradation of fundamental unimodal semantics. Notably, to ensure model compactness and facilitate cross-view knowledge transfer, the Transformer backbones that processing different views share parameters.

**Inference.** Following the multimodal generative paradigm (Zhai et al., 2025), SynGR performs inference via a lightweight autoregressive decoding process. Specifically, given the user behavior history $\mathcal{H}$ represented by sequences of unimodal codebook indices, the model generates the optimal item identifier sequence $\mathbf{Y}$ through beam search. Since the synergistic regularization $\mathcal{L}_{\mathrm{Syn}}$ is exclusively utilized during the training phase to align the representation space, SynGR maintains high efficiency with zero additional computational overhead during inference.

**Complexity Analysis.** The computational complexity of SynGR is asymptotically dominated by the $\mathcal{O}(TN^2d)$ overhead of the Transformer-based backbone. The proposed synergistic components introduce only marginal computational costs: (i) the saliency analysis incurs an $\mathcal{O}(MN^2)$ overhead by repurposing pre-computed attention maps from the forward pass, and (ii) the contrastive alignment adds a negligible $\mathcal{O}(Nd)$ for pooling and similarity computations. Although multi-view training scales the constant factor of the training loss, the inference complexity remains invariant compared to the vanilla Transformer backbone.

*Table 2.* The statistical overview of the three datasets. The "Avg. len." represents the average length of item sequences.

| Datasets | #Users | #Items | #Interactions | Sparsity | Avg. len |
|---|---|---|---|---|---|
| Arts | 22,171 | 9,416 | 174,079 | 99.92% | 7.85 |
| Games | 42,259 | 13,839 | 373,514 | 99.94% | 8.84 |
| Instruments | 17,112 | 6,250 | 136,226 | 99.87% | 7.96 |

# 5. Experiment

In this section, we empirically evaluate the SynGR framework through experiments designed to address five key research questions: **RQ1**: How does SynGR compare against state-of-the-art methods? **RQ2**: What are the training and inference efficiencies of SynGR? **RQ3**: How do different modules affect the performance of SynGR? **RQ4**: How do key parameters influence the model's performance? **RQ5**: How can we intuitively understand the advantage of SynGR?

## 5.1. Experimental Setup

**Datasets.** We evaluate SynGR on three representative categories from the Amazon Product Reviews dataset (Ni et al., 2019): Arts, Games, and Instruments. Detailed statistics for these datasets are provided in Table 2.

**Compared Baselines.** We compare SynGR against three categories of baselines: (i) Sequential Recommendation: BERT4Rec (Sun et al., 2019), SASRec (Kang & McAuley, 2018), FDSA (Zhang et al., 2019), $S^3$-Rec (Zhou et al., 2020), and P5-CID (Geng et al., 2022; Hua et al., 2023); (ii) Multimodal Sequential Recommendation: MISSRec (Wang et al., 2023); (iii) Generative Recommendation: VIP5 (Geng et al., 2023), TIGER (Rajput et al., 2023), MQL4GRec (Zhai et al., 2025), and MACRec (Zhang et al., 2026a). Detailed baseline descriptions are provided in Appendix B.

**Evaluation Metrics.** Following (Zhai et al., 2025; Zhang et al., 2026a), we employ Recall@$K$ and NDCG@$K$ ($K \in \{1, 5, 10\}$) using a leave-one-out protocol. To ensure rigorous evaluation, we perform full ranking over entire item space. Moreover, for all generative models, we standardize the inference process by setting the beam search size to 20.

**Implementation Details.** Due to space limitations, the detailed implementation settings are provided in Appendix C.

## 5.2. Main Experimental Results (RQ1)

To evaluate the effectiveness of SynGR, we conduct comprehensive experiments on three benchmark datasets, with results summarized in Table 1.

SynGR consistently achieves the best performance across all datasets and evaluation metrics, outperforming the strongest multimodal generative baseline, MACRec. Notably, on the Instruments, SynGR surpasses the best baseline by 28.58%

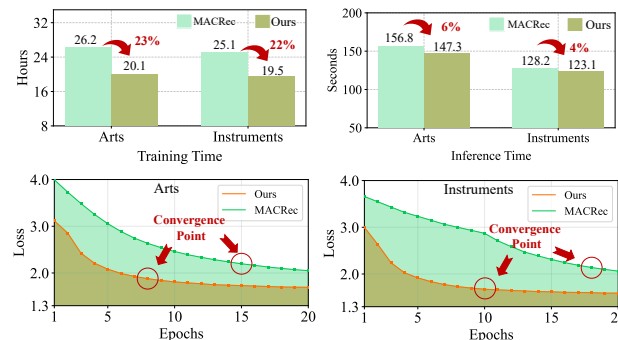

*Figure 3.* Efficiency and convergence analysis of SynGR and MACRec on two datasets. All experiments are conducted on a server equipped with six NVIDIA GeForce RTX 4090 GPUs.

in HR@10 and 12.17% in NDCG@10, highlighting its advantage in scenarios where single-modality cues are insufficient and cross-modal reasoning is critical. These gains are primarily attributed to the saliency-aware masking mechanism, which dynamically identifies and suppresses modality shortcuts, and the synergistic contrastive objective, which regularizes the latent space to prevent unimodal collapse during generation. The consistent improvements across diverse domains demonstrate the robustness of SynGR in capturing complex multimodal semantics for GR.

## 5.3. Efficiency and Convergence Analysis (RQ2)

To assess the practical utility of SynGR, we compare it with the state-of-the-art MACRec in terms of training efficiency, inference latency, and convergence behavior on two datasets.

As shown in Figure 3, SynGR consistently demonstrates advantages in both computational efficiency and optimization stability. Specifically, SynGR reduces training time by approximately 23% on Arts and 22% on Instruments. In contrast, MACRec incurs higher training costs due to its multi-aspect alignment framework, which requires multiple explicit and implicit generation tasks with separate cross-modal constraints. Although SynGR also adopts multi-view objectives, these are processed in parallel within each batch, resulting in a lower per-iteration overhead. This efficiency is further supported by the lightweight design of our modules, where saliency diagnosis reuses existing attention maps with marginal overhead and synergistic contrastive learning incurs negligible additional cost. At inference time, SynGR achieves speedups of 6.0% on Arts and 4.0% on Instruments, highlighting its suitability for real-time deployment. Moreover, SynGR converges faster than MACRec, reaching stable validation loss around epoch 10 on Instruments, whereas MACRec requires nearly 18 epochs. This accelerated convergence suggests suppressing modality shortcuts enables SynGR to capture informative synergistic features early in training, leading to more efficient optimization.

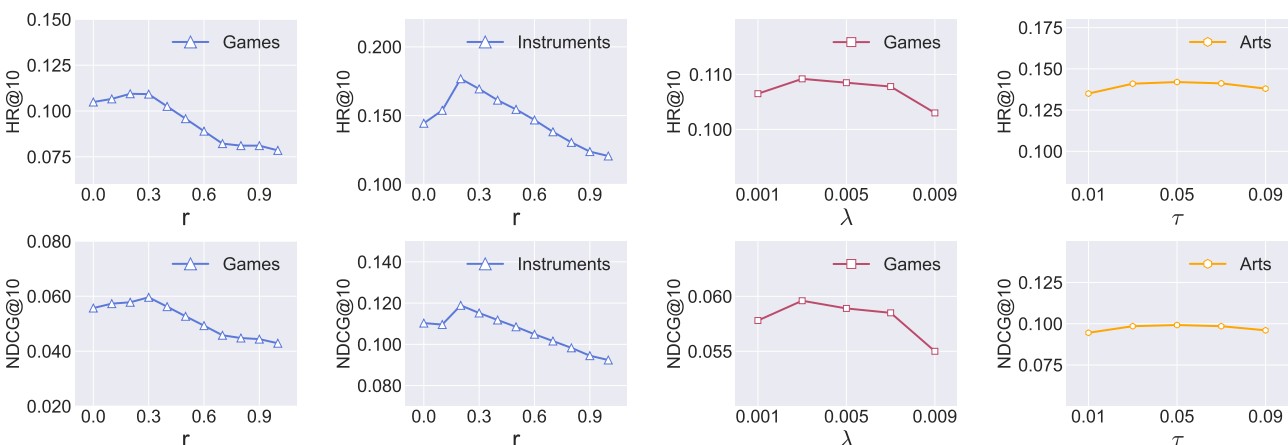

*Figure 4.* The performances (HR@10, NDCG@10) of our SynGR under varying parameters on different datasets.

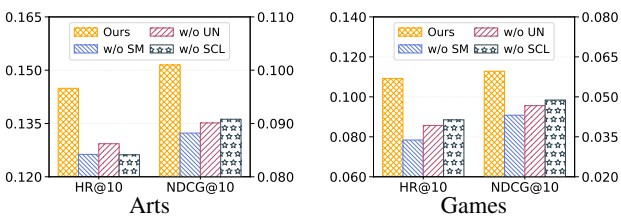

*Figure 5.* Ablation study of SynGR on Arts and Games datasets.

### 5.4. Ablation Study (RQ3)

To validate the effectiveness of the individual components, we conduct an ablation study by comparing the full SynGR model against three specific variants. The implementations of these variants are described as follows:

- **w/o SM**: In this variant, the saliency-based masking strategy is replaced by a random masking protocol to evaluate the importance of targeted feature suppression.
- **w/o UN**: This variant substitutes the constructed unimodal shortcut negative with standard random negative samples from the current training batch to test the impact of shortcut-specific regularization.
- **w/o SCL**: This variant removes the synergistic contrastive learning objective entirely and relies solely on the generative next-token prediction loss.

As illustrated in Figure 5, SynGR consistently achieves the best performance across all metrics on the Arts and Games datasets, validating the necessity of each integrated module. Replacing saliency-based masking strategy with random masking leads to noticeable performance degradation, indicating that unguided perturbations fail to target shortcut features in the dominant modality and thus do not prevent shortcut-driven learning. Removing the unimodal shortcut view construction also degrades performance, suggesting that standard in-batch negatives provide insufficient regularization, while the constructed unimodal view serves as

an effective hard negative that penalizes over-reliance on single-modality redundancy. Moreover, eliminating the synergistic contrastive objective results in further performance drops, particularly on the Games dataset, highlighting that the generative objective alone is insufficient to overcome modality dominance. Overall, these results demonstrate that SynGR benefits from its unified design, which explicitly targets inferential shortcuts and enforces cross-modal interaction to effectively unleash synergistic information.

### 5.5. Parameter Sensitivity Analysis (RQ4)

In this section, we conduct a series of experiments to analyze the sensitivity of SynGR to three key parameters: the masking ratio r in Eq. (11), the weight $\lambda$ of contrastive loss in Eq. (15), and the temperature coefficient $\tau$ in Eq. (13). The corresponding results are visualized in Figure 4.

**Study on Masking Ratio** r. The parameter r controls the proportion of tokens suppressed in the dominant modality to disrupt inferential shortcuts. As shown in the first two columns of Figure 4, performance initially improves as r increases, peaking consistently within the range $[0.2, 0.4]$. This trend corroborates our hypothesis: moderate suppression effectively forces the model to explore cross-modal interactions, thereby mining synergistic semantics. However, extreme masking (r > 0.5) degrades both HR@10 and NDCG@10, indicating that excessive suppression removes essential semantic information. These results underscore the necessity of a balanced adaptive masking strategy.

**Study on Loss Weight** $\lambda$. The parameter $\lambda$ balances the auxiliary synergistic contrastive loss with the primary generative objective. As shown in the third column of Figure 4, performance on the Games dataset improves as $\lambda$ increases from $0.001$, peaking at $\lambda = 0.003$. A small $\lambda$ under-emphasizes synergy extraction, while a large $\lambda$ (> 0.005) causes performance to drop, suggesting that ex-

*Figure 6.* Qualitative comparison between SynGR and MACRec. Given the same user history, SynGR ranks the ground-truth (GT) item at the top position, whereas MACRec ranks it only third, favoring items with coarse similarity to football- or jersey-related concepts.

cessive regularization interferes with the primary task.

**Study on Temperature Coefficient $\tau$.** Finally, we examine the impact of the temperature $\tau$ in the InfoNCE loss, which controls the sharpness of the latent space distribution. As shown in the rightmost column of Figure 4, performance remains stable across a reasonable range, peaking when $\tau$ is between $0.05$ and $0.07$. This suggests the learned representations are robust, with the contrastive objective effectively distinguishing between synergy-critical views and unimodal hard negatives without extensive hyperparameter tuning. More parameter studies are provided in Appendix D.

### 5.6. Case Study (RQ5)

To provide a more intuitive understanding of the superiority of SynGR, we conduct a case study by comparing the recommendation results of SynGR and the strongest baseline MACRec. As shown in Figure 6, given a user history containing a 2022 World Cup match ball, an Argentina World Cup jersey, gold boots, and a Messi World Cup poster, the ground-truth next item is a Messi World Cup Winner Poster. This recommendation requires integrating textual cues such as "Messi", "Argentina", and "World Cup" with visual cues such as jersey style, gold-themed football elements, and poster-like layouts, rather than relying on a single modality.

SynGR successfully ranks the ground-truth item as the top recommendation, while MACRec places a Molten match ball and a Cristiano Ronaldo jersey ahead of it. This indicates that MACRec tends to focus on dominant unimodal similarities, such as football appearance or jersey-related semantics, whereas SynGR better captures the cross-modal synergistic intent centered on "Messi winning the World Cup". This case provides qualitative evidence that the pro-

posed saliency-aware masking and synergistic contrastive learning mechanisms can effectively suppress unimodal shortcuts and encourage the model to discover more discriminative cross-modal synergistic information.

## 6. Conclusion

In this paper, we studied the role of synergistic information in multimodal GR and identified a synergy gap caused by shortcut-driven optimization under generative objectives. To address this problem, we proposed SynGR, a synergistic generative recommendation framework that disrupts unimodal shortcuts and promotes the use of cross-modal dependencies during generation. Experiments on three datasets showed SynGR consistently improves recommendation performance, achieving an average gain of approximately 10%.

Although SynGR takes an initial step toward modeling modal synergy for GR, there is room for improvement. First, the current framework focuses primarily on discovering item-side synergistic semantics, incorporating interaction-driven collaborative signals could further enhance personalized recommendation. Additionally, extending the paradigm to richer modality combinations, such as audio and video, would be an interesting direction for testing its broader applicability across diverse recommendation scenarios.

## Acknowledgments

Supported by the National Key R&D Program of China (Grant No. 2024YFF0729003), NSFC (Nos. 62176014, 62276015, 62206266), the Fundamental Research Funds for the Central Universities, and the Academic Excellence Foundation of BUAA for PhD Students.

## Impact Statement

This work focuses on the technical aspects of generative recommendation and does not have direct negative societal implications. The proposed method is evaluated on standard public datasets and does not involve the use of personal information or controversial applications of generative AI.

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

# A. Proof of Lemma 1

Let $(\mathbf{X_v}, \mathbf{X_t})$ denote the multimodal input consisting of visual and textual histories, $\mathbf{Y}$ be the target item identifier, and $\widehat{\mathbf{X}} = \phi(\mathbf{X_v}, \mathbf{X_t})$ be a synergy-preserving view as defined in Definition 1. We consider the following Markov chain:

$$(\mathbf{X_v}, \mathbf{X_t}) \xrightarrow{\phi} \widehat{\mathbf{X}} \xrightarrow{\text{Transformer}} \mathbf{Z}_{\text{syn}} \xrightarrow{\text{Predictor}} \mathbf{Y}. \tag{16}$$

Intuitively, $\mathbf{Z}_{\text{syn}}$ is expected to remain predictive of $\mathbf{Y}$ while avoiding reliance on information that can be recovered from either modality alone. We show that, under this construction, the resulting representation is dominated by synergistic information under the PID framework.

Under the Partial Information Decomposition (PID) framework (Tax et al., 2017), the joint mutual information between the two sources and the target can be decomposed into four non-negative components:

$$\mathrm{I}(\mathbf{X_v}, \mathbf{X_t}; \mathbf{Y}) = \mathbf{R} + \mathbf{U_v} + \mathbf{U_t} + \mathbf{S}. \tag{17}$$

Here, $\mathbf{R}$ denotes the information about $\mathbf{Y}$ redundantly available from either modality, $\mathbf{U_v}$ and $\mathbf{U_t}$ denote the modality-specific unique information contributed by the visual and textual streams respectively, and $\mathbf{S}$ captures the synergistic information that emerges only through their joint observation.

We begin by establishing an upper bound on the predictive information carried by $\mathbf{Z}_{\text{syn}}$. From the Markov chain in Eq. (16), the Data Processing Inequality (DPI) (Beaudry & Renner, 2011) implies that any representation derived from $\widehat{\mathbf{X}}$ cannot contain more information about $\mathbf{Y}$ than $\widehat{\mathbf{X}}$ itself:

$$\mathrm{I}(\mathbf{Z}_{\text{syn}}; \mathbf{Y}) \leq \mathrm{I}(\widehat{\mathbf{X}}; \mathbf{Y}). \tag{18}$$

Moreover, by the Joint Sufficiency property of $\phi$ (Definition 1), the transformation $\widehat{\mathbf{X}}$ preserves, up to approximation, all task-relevant information in the original multimodal input:

$$\mathrm{I}(\widehat{\mathbf{X}}; \mathbf{Y}) \approx \mathrm{I}(\mathbf{X_v}, \mathbf{X_t}; \mathbf{Y}). \tag{19}$$

Combining Eq. (18) and Eq. (19), we obtain

$$\mathrm{I}(\mathbf{Z}_{\text{syn}}; \mathbf{Y}) \lesssim \mathbf{R} + \mathbf{U_v} + \mathbf{U_t} + \mathbf{S}, \tag{20}$$

where the decomposition follows directly from Eq. (17).

We now examine how the unimodal obscuration constraints shape the composition of $\mathbf{Z}_{\text{syn}}$. The central idea is to prevent $\mathbf{Z}_{\text{syn}}$ from encoding information that is already accessible from either unimodal stream alone. A natural and widely adopted way to express this requirement is to enforce small mutual information between $\mathbf{Z}_{\text{syn}}$ and each modality:

$$\mathrm{I}(\mathbf{Z}_{\text{syn}}; \mathbf{X_v}) \leq \epsilon, \qquad \mathrm{I}(\mathbf{Z}_{\text{syn}}; \mathbf{X_t}) \leq \epsilon, \tag{21}$$

for a small $\epsilon > 0$. Such constraints are standard in information bottleneck and invariance-based learning, where the goal is to retain task-relevant information while suppressing shortcut or nuisance factors.

Under the PID semantics, any information about $\mathbf{Y}$ that is accessible from $\mathbf{X_v}$ alone must belong to either the redundant component $\mathbf{R}$ or the visual-unique component $\mathbf{U_v}$, since

$$\mathrm{I}(\mathbf{X_v}; \mathbf{Y}) = \mathbf{R} + \mathbf{U_v}. \tag{22}$$

An analogous relation holds for the textual modality:

$$\mathrm{I}(\mathbf{X_t}; \mathbf{Y}) = \mathbf{R} + \mathbf{U_t}. \tag{23}$$

Therefore, the constraints in Eq. (21) explicitly limit the extent to which $\mathbf{Z}_{\text{syn}}$ can encode information attributable to $\mathbf{R}$, $\mathbf{U_v}$, or $\mathbf{U_t}$, all of which are individually recoverable from at least one modality. As a result, up to an approximation controlled by $\epsilon$, the only $\mathbf{Y}$-relevant information that can be consistently preserved in $\mathbf{Z}_{\text{syn}}$ is the portion that is not accessible from either modality in isolation, namely, the synergistic component $\mathbf{S}$.

Taken together, Eq. (20) and Eq. (21) yield the following interpretation: the learning objective encourages $\mathbf{Z}_{\mathrm{syn}}$ to remain predictive of $\mathbf{Y}$ through Joint Sufficiency, while the unimodal obscuration constraints systematically suppress shortcut-dominant information. Under the PID decomposition, the only component that satisfies both requirements is the synergistic term. Consequently, at the optimum (or in the limit $\epsilon \to 0$), $\mathbf{Z}_{\mathrm{syn}}$ is dominated by synergistic information:

$$\mathrm{I}(\mathbf{Z}_{\mathrm{syn}}; \mathbf{Y}) \approx \mathbf{S}. \tag{24}$$

This completes the proof. $\square$

**Remark.** We stress that Eq. (24) should be interpreted as an inductive-bias characterization rather than a strict finite-sample guarantee. In practical high-dimensional settings, exact elimination of PID components is generally infeasible; instead, the proposed objective and constraints bias the optimization toward representations whose predictive power primarily arises from cross-modal interaction, as corroborated by empirical evidence (Liang et al., 2021; Ye et al., 2022).

## B. Baselines Details

We compare SynGR with the following representative baseline methods:

- **GRU4Rec** (Hidasi et al., 2016) pioneers the application of Recurrent Neural Networks (RNNs) with Gated Recurrent Units to capture the dynamic evolution of user interests within session-based data.

- **SASRec** (Kang & McAuley, 2018) applies a self-attention mechanism within a unidirectional sequential model to identify relevant items from a user's historical interactions.

- **BERT4Rec** (Sun et al., 2019) adapts the deep bidirectional self-attention architecture of BERT to recommendation tasks, utilizing a Cloze objective to predict masked items based on both left and right contexts.

- **FDSA** (Zhang et al., 2019) incorporates context features into sequential recommendation by employing two distinct self-attention blocks to capture transitions at both the item and feature levels.

- **S³-Rec** (Zhou et al., 2020) is a pre-training framework based on self-supervised learning that leverages mutual information maximization to model the correlations between items, attributes, and subsequences.

- **VQ-Rec** (Hou et al., 2023) employs a vector quantization mechanism to map items into discrete codes, facilitating effective knowledge transfer across different domains by learning transferable item representations.

- **MISSRec** (Wang et al., 2023) proposes a transfer learning-based framework designed to effectively map multimodal features (visual and textual) into the sequential recommendation process.

- **P5-CID** (Geng et al., 2022; Hua et al., 2023) unifies various recommendation tasks into a shared text-to-text framework based on the T5 architecture, treating recommendation essentially as a natural language generation problem. We report results for the P5-CID variant.

- **VIP5** (Geng et al., 2023) extends the P5 paradigm by integrating visual and textual modalities into a foundation model structure to enhance personalization capabilities through multi-modal understanding.

- **TIGER** (Rajput et al., 2023) introduces a generative retrieval approach that utilizes RQ-VAE to assign unique hierarchical semantic IDs to items and trains a Transformer to generate these identifiers auto-regressively.

- **MQL4GRec** (Zhai et al., 2025) presents a method to transform multimodal information into a discrete quantized language, allowing the generative model to effectively utilize rich side information during the recommendation process.

- **MACRec** (Zhang et al., 2026a) stands as the current state-of-the-art generative model, which constructs superior semantic IDs through a multi-aspect cross-modal quantization framework to align diverse modalities.

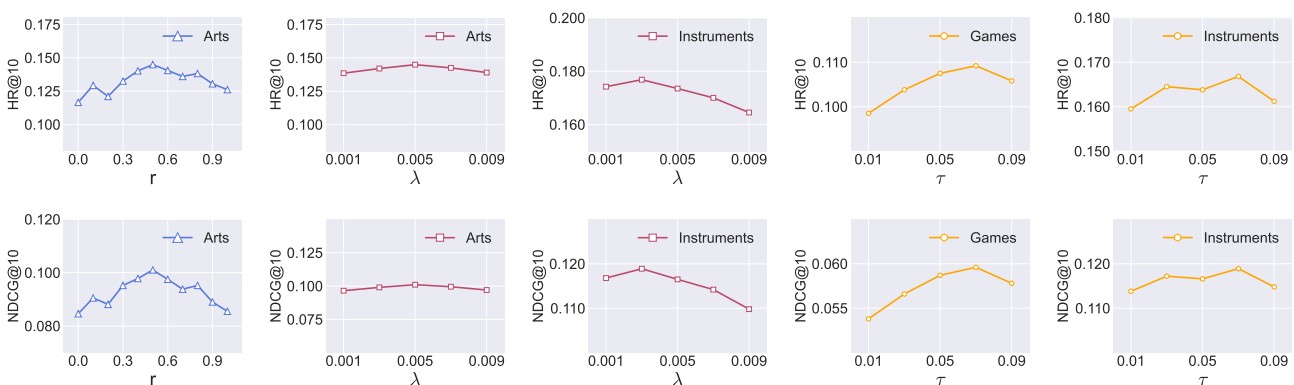

*Figure 7.* Additional sensitivity analysis on remaining datasets.

*Table 3.* Pre-training dataset statistics.

| Dataset | Scale | | | Characteristics | |
| --- | --- | --- | --- | --- | --- |
| | **Users** | **Items** | **Interactions** | **Sparsity** | **Avg. Len** |
| Pet Supplies | 183,697 | 31,986 | 1,571,284 | 99.97% | 8.55 |
| Cell Phones | 123,885 | 38,298 | 873,966 | 99.98% | 7.05 |
| Automotive | 105,490 | 39,537 | 845,454 | 99.98% | 8.01 |
| Tools | 144,326 | 41,482 | 1,153,959 | 99.98% | 8.00 |
| Toys | 135,748 | 47,520 | 1,158,602 | 99.98% | 8.53 |
| Sports | 191,920 | 56,395 | 1,504,646 | 99.99% | 7.84 |

*Algorithm 1.* Normalized PID Component Computation

**Input:** Metric evaluation function $M(\cdot)$
Ground-truth targets $\mathbf{Y}_{\text{true}}$,
Predictions: text-only $\mathbf{Y}_t$, vision-only $\mathbf{Y}_v$,
joint bi-modal $\mathbf{Y}_j$
**Output:** Normalized PID components: $\mathbf{S}, \mathbf{R}, \mathbf{U_t}, \mathbf{U_v}$
1: $P_t \leftarrow M(\mathbf{Y}_t, \mathbf{Y}_{\text{true}}), P_v \leftarrow M(\mathbf{Y}_v, \mathbf{Y}_{\text{true}})$
2: $P_j \leftarrow M(\mathbf{Y}_j, \mathbf{Y}_{\text{true}})$
3: $\mathbf{S} \leftarrow \frac{\max(0,\ P_j - \max(P_t, P_v))}{P_j}, \mathbf{R} \leftarrow \frac{\min(P_t, P_v)}{P_j}$
4: $\mathbf{U_t} \leftarrow \frac{P_t - \min(P_t, P_v)}{P_j}, \mathbf{U_v} \leftarrow \frac{P_v - \min(P_t, P_v)}{P_j}$
5: **return** $\mathbf{S}, \mathbf{R}, \mathbf{U_t}, \mathbf{U_v}$

## C. Implementation Details

Following the protocols in (Zhai et al., 2025; Zhang et al., 2026a), we adopt a two-stage training paradigm. SynGR is first pre-trained on a massive multimodal corpus derived from six Amazon categories: Pet Supplies, Cell Phones, Automotive, Tools, Toys, and Sports. This corpus comprises approximately 7.1M interactions among 884,918 users and 255,181 items, providing a robust foundation for cross-modal semantic alignment. The detailed statistics of these pre-training datasets are summarized in Table 3. After pre-training, the model is fine-tuned on specific downstream target domains.

We employ T5 (Raffel et al., 2020) as our generative backbone. Both the encoder and decoder consist of a 4-layer Transformer structure, with 6 self-attention heads and a hidden dimension of $d = 64$ per layer. For feature extraction, we utilize LLaMA for textual semantics and ViT-L/14 for visual representations. The RQ-VAE module is configured with a codebook size of $M = 256$ and 4 quantization layers to discretize multimodal content into hierarchical identifiers.

The framework is optimized using AdamW (Li et al., 2025c; Zhou et al., 2022; 2026). We fix the batch size to 1024 to ensure stable large-scale training and set the contrastive temperature $\tau$ to 0.07 for the synergistic objective throughout all experiments. The learning rate is selected from $\{1e\text{-}4, 5e\text{-}4, 1e\text{-}3\}$, weight decay from $\{0, 0.01, 0.1\}$, mask ratio from $\{0, 0.1, \dots, 1.0\}$, warmup ratio from $\{0, 0.05, 0.1\}$, contrastive weight from $\{0.001, 0.003, 0.005, 0.007, 0.009\}$, and batch size from $\{512, 1024\}$, with the best configuration selected based on validation performance. The feature dimension is fixed to 256 and all models are trained for 200 epochs. The calculation details of the PID are shown in Algorithm 1.

## D. Parameter Sensitivity Analysis

Additional sensitivity results are reported in Figure 7.

