# OpenReview forum: "SynGR: Unleashing the Potential of Cross-Modal Synergy for Generative Recommendation"
_ICML.cc/2026/Conference — ICML 2026 regular_

### Official Review · Reviewer_taut · 2026-03-09

**Soundness:** 4
**Presentation:** 4
**Significance:** 3
**Originality:** 3
**Overall Recommendation:** 5
**Confidence:** 4

**Summary:**

The paper proposes SynGR, a novel synergistic generative recommendation framework designed to move beyond simple alignment-centric fusion in multimodal GR. The authors argue that existing methods primarily focus on modality agreement, failing to capture “synergistic information”, emergent item properties that can only be inferred through the joint interplay of multiple modalities. SynGR addresses this by explicitly encouraging the exploitation of cross-modal dependencies and constraining over-reliance on any single dominant modality. Extensive experiments across three benchmark datasets show an average performance improvement of 9.01%.

**Compliance With Llm Reviewing Policy:**

Affirmed.

**Final Justification:**

The authors have fully addressed my concerns.

**Key Questions For Authors:**

Please see above weaknesses.

**Limitations:**

The paper still has several limitations. It mainly focuses on positive cross-modal synergy, while providing limited discussion of how conflicting signals or negative synergy are handled when modalities are noisy or inconsistent. In addition, the scalability of the proposed synergistic objective to more complex multimodal settings beyond text-image pairs, such as video, audio, or knowledge graphs, remains unclear. Finally, although the paper highlights synergistic information, the roles and effective use of redundant and modality-unique information are not sufficiently explored.

**Strengths And Weaknesses:**

Strengths
1.	This paper is well-written and the proposed SynGR method is novel. Some concepts, such as PID and cross-modal synergy, with intuitive narrative flow that is easy to follow.
2.	The proposed method leverages Partial Information Decomposition (PID) theory to offer theoretical guarantees for multimodal synergy.
3.	Extensive experiments on three datasets demonstrate a substantial average improvement of 9.01%, solidly validating the method's superiority.
4.	The paper provides a comprehensive ablation study that effectively demonstrates the necessity of various key design choices in SynGR.

Weaknesses:
1.	The paper focuses primarily on positive synergy but lacks a detailed discussion on how the model handles “negative synergy” or noise when modalities provide conflicting information.
2.	It remains unclear how the synergistic objective scales beyond bimodal (text/image) data to more complex scenarios involving video, audio, or structured knowledge graphs.
3.	While the paper emphasizes the importance of capturing synergistic information, the specific roles and utilization strategies for redundant and unique information remain somewhat under-explored.

---

> ### Author Rebuttal · Authors · 2026-03-30
>
> We are grateful for your efforts in reviewing the paper. Below, we do our utmost to address your concerns.
>
> >**Q1: Lacks a detailed discussion on how SynGR handles “negative synergy” or noise.**
>
> Thanks!
>
> (1) First, our synergistic objective is conditioned on the target item $\mathbf{Y}$. Under the PID framework, if one modality contains conflicting noise, its task-relevant mutual information with the target becomes small. In this case, the Saliency-Aware Masking mechanism will down-weight the noisy modality, while the contrastive objective discourages the model from relying on such noisy unimodal shortcuts.
>  Therefore, SynGR **does not force the fusion of conflicting signals**, but instead preserves informative cross-modal evidence and suppresses noise.
>
>  (2) Second, we also verified this **empirically through a Modality Perturbation Test** by injecting Gaussian noise into the embeddings of 10\%–90\% of items in Instruments dataset. As shown in **Table 1**, MACRec exhibits a sharper performance decline as the noise level increases, whereas SynGR shows a **more gradual degradation** trend. This result demonstrates that SynGR can effectively mitigate negative synergy under conflicting multimodal conditions.
>
> ---Table 1---
>
> | Model | Metric | 10% | 30% | 50% | 70% | 90% |
> |-------|--------|-----|-----|-----|-----|-----|
> | MACRec | H@10 | 0.1236 | 0.0581 | 0.0223 | 0.0149 | 0.0004 |
> | SynGR | H@10 | **0.1329** | **0.0735** | **0.0396** | **0.0176** | **0.0009** |
> | MACRec | N@10 | 0.0926 | 0.0439 | 0.0172 | 0.0010 | 0.0001 |
> | SynGR | N@10 | **0.1031** | **0.0672** | **0.0372** | **0.0121** | **0.0004** |
>
>
> >**Q2: It remains unclear how the synergistic objective scales beyond bimodal (text/image) data to more complex scenarios.**
>
> Thanks! SynGR is inherently modality-agnostic and naturally scales to handle more than two modalities:
>
> (1) **Theoretical**: SynGR maps any continuous modality into a unified discrete token sequence. To scale to multiple modalities, the masking mechanism evaluates attention scores across all candidate modalities to mask the dominant one. Similarly, the contrastive learning objective seamlessly extends by constructing unimodal negative views for each additional modality, explicitly penalizing any single-modality shortcut.
>
> (2) **Empirical**: To demonstrate this, we extended SynGR to incorporate structural collaborative signals (user-item graph) as a third modality. We extracted structural embeddings via a graph convolutional network, quantized them into discrete tokens, and integrated them alongside the textual and visual tokens.
> As shown in **Table 2**, integrating this third modality (user-item graph) consistently improves recommendation performance, proving that SynGR can effectively extract higher-order synergistic information from complex, multi-modal scenarios.
>
> ---Table 2---
>
> | Dataset    | Model                        | H@10  | N@10 |
> |------------|------------------------------|--------|---------|
> | Instruments| SynGR (2 Modalities)          | 0.1768 | 0.1189  |
> | Instruments| SynGR (3 Modalities) | **0.1803** | **0.1284** |
> | Arts       | SynGR (2 Modalities )      | 0.1449 | 0.1010  |
> | Arts       | SynGR (3 Modalities) | **0.1478** | **0.1030** |
>
> >**Q3: The specific roles and utilization strategies for redundant and unique information remain somewhat under-explored.**
>
> Thanks!
>
> (1) We emphasize synergistic information ($\mathbf{S}$) in our paper primarily because it is a critical signal that existing models consistently overlook, not because redundant ($\mathbf{R}$) and unique ($\mathbf{U}$) information are unimportant. On the contrary, $\mathbf{R}$ and $\mathbf{U}$ serve as the indispensable semantic foundation of our multimodal recommendation framework. In fact, our model implicitly but firmly preserves both $\mathbf{R}$ and $\mathbf{U}$ within the generative loss $\mathcal{L}_{Gen}$.
>
> (2) To **explicitly quantify** their vital roles, we conducted an ablation study on the Instruments dataset by removing the generative objectives corresponding to redundant information (w/o $\mathbf{R}$), unique information (w/o $\mathbf{U}$), and both (w/o $\mathbf{UR})$. As shown in **Table 3**, discarding either $\mathbf{R}$ or $\mathbf{U}$ leads to a significant performance drop, and removing both results in severe degradation. This empirically confirms that $\mathbf{R}$ and $\mathbf{U}$ form the essential semantic baseline, upon which our synergistic module builds to achieve state-of-the-art performance.
>
> ---Table 3---
>
> | Model Variant | H@10   | N@10 |
> |---------------|---------|---------|
> | w/o $\mathbf{R}$     | 0.1272  | 0.0984  |
> | w/o $\mathbf{U}$     | 0.1125  | 0.0882  |
> | w/o $\mathbf{UR}$    | 0.0557  | 0.0473  |
> | SynGR   | **0.1768**  | **0.1189**  |

---

> > ### Author Rebuttal · Reviewer_taut · 2026-04-03
> >
> > The authors have fully addressed my concerns.

---

> > > ### Author Response · Authors · 2026-04-04
> > >
> > > We sincerely appreciate your valuable reviews and positive feedback!
> > >
> > > We hope the idea and approach presented in this work can inspire more studies in this direction.

---

### Official Review · Reviewer_1Sqh · 2026-03-12

**Soundness:** 3
**Presentation:** 3
**Significance:** 3
**Originality:** 2
**Overall Recommendation:** 4
**Confidence:** 2

**Summary:**

This paper proposes SynGR, a framework designed to address the insufficient utilization of multi-modal synergistic information in Generative Recommendation (GR). By leveraging Partial Information Decomposition (PID) theory, the authors analyze the limitations of existing methods and introduce the SynGR framework, which incorporates Saliency-Aware Masking and Synergistic Contrastive Learning. Experiments conducted on three Amazon datasets demonstrate that this method significantly outperforms existing state-of-the-art baselines.

**Compliance With Llm Reviewing Policy:**

Affirmed.

**Key Questions For Authors:**

See weaknesses.

**Strengths And Weaknesses:**

Strengths

+ Empirical analysis (Fig. 1) and theoretical derivations based on Partial Information Decomposition (PID) demonstrate that existing alignment-centric methods fail to capture emergent semantics.
+ The paper formalizes the problem from an information-theoretic perspective using the PID framework and designs specific algorithmic modules accordingly; notably, the theoretical derivation (Lemma 1 and its proof) provides a solid logical foundation for the method's effectiveness.
+ Efficiency analysis reveals that SynGR not only converges faster but also achieves superior training and inference times compared to MACRec, validating the efficacy of its lightweight design.

Weaknesses

*   The synergistic contrastive learning module constructs unimodal negative samples, with losses calculated symmetrically for text and visual sub-tasks. However, it remains unclear how to ensure in practice that these unimodal views genuinely represent "shortcuts" rather than becoming invalid inputs due to critical information loss. Specifically regarding the visual modality, is using visual tokens alone sufficient to constitute a meaningful negative sample, or does it merely act as a weakened positive sample?
*   The current experiments are limited to three product categories from Amazon datasets. Given the significant variations in image-text complementarity across different domains (e.g., visuals are paramount for fashion, while text is crucial for books), the generalizability of SynGR under diverse data distributions remains an open question.
*   The abstract claims an "average improvement of 9.01%," yet the specific metric gains reported in the main body vary widely, ranging from 1.30% to 28.58%. It is necessary to clarify the calculation method for this average and discuss whether such aggregated figure meaningfully reflects the model's performance given the high variance.

---

> ### Author Rebuttal · Authors · 2026-03-30
>
> Sincerely thanks for your efforts in reviewing the paper. Below, we respond to your questions in detail.
>
> >**Q1: About the selection of negative samples.**
>
> We sincerely appreciate the reviewer’s insightful question!
>
> (1) To examine whether the unimodal view acts as a weakened positive signal or a shortcut, we analyzed modality contribution weights across datasets (**Table 1**). As shown, the dominant modality varies across domains (e.g., text dominates in Instruments and Games, while visual features dominate in Arts). This suggests that, depending on the scenario, models tend to rely on the dominant modality. Such reliance may function as a shortcut that the model preferentially exploits, rather than merely representing a weakened or invalid input.
>
> ---Table 1---
>
> | Modality | Instruments | Arts | Games |
> | :--- | :---: | :---: | :---: |
> | Text | **0.5943** | 0.4752 | **0.6075** |
> | Visual | 0.4057 | **0.5248** | 0.3925 |
>
> (2) From an information-theoretic perspective, a unimodal visual view mainly provides redundancy and uniqueness $(\mathbf{R} + \mathbf{U}_v)$, but lacks cross-modal synergy. Since GR models tend to follow the path of least resistance, relying on $(\mathbf{R} + \mathbf{U}_v)$ can partially minimize the generation loss while limiting the learning of cross-modal alignment. In this sense, the unimodal view may act as an optimization shortcut that hinders the emergence of synergistic information $\mathbf{S}$.
>
> (3) Finally, treating this unimodal shortcut as a negative sample does not force the model to discard visual information. Our primary generative objective ($\mathcal{L}_{Gen}$) already guarantees the preservation of $(\mathbf{R} + \mathbf{U}_v)$. The contrastive loss acts as a regularizer, pushing the representation out of this unimodal trap to compel the **extraction of the missing synergy** ($\mathbf{S}$).
>
>
> >**Q2: The generalizability of SynGR under diverse data distributions.**
>
> Thanks! We **conducted extensive experiments** on two massive, domain-distinct datasets:
>
> ML-25M: A profoundly text-dominant domain (~25 million interactions), where recommendations rely heavily on textual metadata (e.g., movie titles, genres, plot summaries).
>
> Taobao: A profoundly vision-dominant domain (~85 million interactions), consisting of e-commerce items where visual aesthetics are the primary driving factor.
>
> The performance comparison with the strongest baselines (MQL4Rec and MACRec) is summarized below:
>
> ---Table 2---
>
> | Method  | ML-25M (H@10) | ML-25M (N@10) | Taobao (H@10) | Taobao (N@10) |
> |---------|---------------|---------------|---------------|---------------|
> | MQL4Rec | 0.0803        | 0.2500        | 0.0609        | 0.2311        |
> | MACRec   | 0.0702        | 0.2455        | 0.0615        | 0.2325        |
> | SynGR  | **0.0856** (+6.6%)        | **0.2714** (+8.6%)       | **0.0696** (+14.3%)        | **0.2584** (+11.1%)        |
>
> As the results demonstrate, SynGR consistently outperforms all baselines in both text-dominant and vision-dominant scenarios. This empirically validates that our method is not overfitted to a specific modality ratio, showcasing strong generalizability under diverse data distributions.
>
> >**Q3: About the calculation method for 9.01.**
>
> Thanks!
>
>  (1) First, the 9.01% is the **arithmetic mean** of all relative improvements over the best baselines across all metrics and datasets reported in Table 1.
>
>  (2) Second, for domain-specific items where unimodal shortcuts are **largely sufficient** (e.g., matching distinct franchise titles or brands in the Games dataset), the "synergy gap" is naturally small. Consequently, baseline models already perform adequately, yielding modest relative gains for SynGR (e.g., +1.30% on HR@10).
>
> Conversely, for complex items that **require holistic cross-modal semantic interaction** (e.g., the intricate visual aesthetics and textual specifications in the Instruments dataset), existing baseline models fail due to unimodal collapse. Here, SynGR explicitly recovers the critical synergistic information (S), yielding massive relative improvements (e.g., +28.58% on HR@10).
>
>
> (3) Additionally, we evaluated our model on two new massive-scale datasets (**Table 2**), where SynGR continues to demonstrate robust performance
>
> Thus, the variance proves that SynGR **precisely targets** and resolves the synergy bottleneck rather than applying a generic, uniform scaling.

---

> > ### Author Rebuttal · Reviewer_1Sqh · 2026-04-02
> >
> > My concerns have been addressed.

---

> > > ### Author Response · Authors · 2026-04-04
> > >
> > > Thank you for your thoughtful feedback and for acknowledging that the concerns have been addressed.
> > >
> > > Given the revisions and additional clarifications provided, we kindly request that you **reconsider the score** in light of these improvements.

---

### Official Review · Reviewer_nPWh · 2026-03-12

**Soundness:** 3
**Presentation:** 2
**Significance:** 3
**Originality:** 2
**Overall Recommendation:** 3
**Confidence:** 3

**Summary:**

This paper proposes to prevent recommendation models from collapsing back to single-modal representations by leveraging synergistic information across modalities, thereby improving recommendation performance. The authors introduce a Saliency-Aware Masking Mechanism and Synergistic-Aware Contrastive Learning, which masks out tokens from over-attended modalities during sequence reconstruction to enable the model to learn synergistic information. The method achieves some improvements on several Amazon datasets.

**Compliance With Llm Reviewing Policy:**

Affirmed.

**Key Questions For Authors:**

Is there a before-and-after comparison to verify whether modal collapse is indeed prevented? Like Figure 1C

The backbone is the same as MacRec, but MacRec does not use pre-training. How significant is the impact of pre-training on final performance of your model?

Multimodal recommendation methods should theoretically improve long-tail item more significantly. Does this method potentially harm recommendations for long-tail items?

MQLRec demonstrates zero-shot capability. Does SynGR possess similar capabilities?

**Limitations:**

The model is based on the same backbone as MacRec, yet shows substantial improvements. Since the proposed method appears to be a general approach, could the authors validate its effectiveness on additional backbones?

**Strengths And Weaknesses:**

# Strengths:

Demonstrates the importance of synergistic information, not through modal alignment but by extracting synergistic information itself.

Proposes a masking approach to prevent collapse into single-modal representations.

The loss design is somewhat interesting: by removing tokens with the highest attention scores while still reconstructing the original sequence, it effectively prevents over-attention to a specific modality.

# Weaknesses:
The writing quality needs improvement.

The method appears incremental; the main contribution is the masking strategy, which is rather straightforward and lacks deeper insights.

Experiments are conducted only on Amazon datasets. Could the authors validate the method on larger-scale datasets? Moreover, the evaluation only uses head-focused metrics (HR@1, HR@5, HR@10, N@5, N@10).

---

> ### Author Rebuttal · Authors · 2026-03-30
>
> Thanks!
>
> >**Q1: About writing quality.**
>
> We appreciate your feedback and will conduct a thorough proofreading of the final version.
>
> We welcome **any further discussion** during the rebuttal phase.
>
> >**Q2: The method appears incremental.**
>
> In fact, our main contribution **does not lie in the masking operation** itself, but in being the first to use PID to reveal the fundamental unimodal shortcut in GR. Existing GR models may rely on this shortcut to reduce the training objective, while overlooking critical synergistic information.
>
> SynGR is then introduced as a **lightweight and targeted** solution to this information-theoretic issue. Its effectiveness is supported by strong empirical results, suggesting that it is **not an incremental** tweak but a necessary paradigm shift to capture true synergy.
>
>
> >**Q3: Larger-scale datasets and broader metrics.**
>
> (1) We further **conducted experiments**  on two large-scale datasets with diverse data distributions: ML-25M (approx. 25 million interactions, where text is often crucial) and Taobao (approx. 85 million interactions, heavily visual-driven). As summarized below, SynGR demonstrates highly competitive generalizability across both domains, maintaining robust performance.
>
> | Method  | M(H@10) | M (N@10) | T (H@10) | T(N@10) |
> |---------|---------------|---------------|---------------|---------------|
> | MQL4Rec | 0.0803        | 0.2500        | 0.0609        | 0.2311        |
> | MACRec   | 0.0702        | 0.2455        | 0.0615        | 0.2325        |
> | SynGR  | **0.0856**       | **0.2714**       | **0.0696**        | **0.2584**       |
>
>
> (2) We also evaluated **additional metrics** at @30 and @50. The results further demonstrate SynGR's superiority.
>
> | Model | Dataset | N@30 | N@50 |
> | :--- | :--- | :---: | :---: |
> | MACRec | Arts | 0.0985 | 0.1052 |
> | SynGR | Arts | **0.1091** | **0.1164** |
> | MACRec | Games | 0.0631 | 0.0655 |
> | SynGR | Games | **0.0714** | **0.0738** |
>
> >**Q4: Verify whether modal collapse is prevented.**
>
> | Method |  $\mathbf{S}$ | $\mathbf{R}$ |  $\mathbf{U}_t$ |  $\mathbf{U}_v$ |
> | :--- | :---: | :---: | :---: | :---: |
> | MQL4Rec | 6.4% | 63.5% | 20.2% | 9.9% |
> | MACRec | 7.5% | 60.0% | 20.0% | 12.5% |
> | SynGR | **29.1%** | 45.0% | 16.0% | 9.9% |
>
> Empirical PID analysis reveals that baselines collapse into unimodal shortcuts with critically low synergy. SynGR effectively prevents this modal collapse by forcing the extraction of cross-modal semantics.
>
> >**Q5: Impact of pre-training.**
>
> (1) SynGR and MACRec **share the same pre-training** phase. Therefore, the performance gains of SynGR can be attributed to the proposed synergy-extraction mechanism rather than any pre-training advantage.
>
> (2) To further assess the role of pre-training, we **conducted an ablation study** comparing SynGR with and without pre-training on Instruments (I) and Arts (A).
>
> | Model Variant | I(H@10) | I(N@10) | A(H@10) | A (N@10) |
> |---------------|---------------------|-----------------------|--------------|----------------|
> | w/o pretrain | 0.1462 | 0.1053 | 0.1308 | 0.0941 |
> | SynGR | **0.1768** | **0.1189** | **0.1449** | **0.1010** |
>
> As shown above, pre-training provides a **beneficial initialization** that improves the final performance.
>
> >**Q6: Performance for long-tail items.**
>
> Thanks! SynGR **does not harm** long-tail items; rather, it **improves** them. Unlike head items, which often contain strong unimodal signals, long-tail items rely more on cross-modal synergistic information.
> By disrupting unimodal shortcuts, SynGR better captures such synergy.
>
> Empirically, we conducted **additional experiments** on Arts and found SynGR achieves an N@10 of 0.0142 on the bottom 50% tail items, outperforming MACRec (0.0111).
>
> >**Q7: Zero-shot capability.**
>
> Yes, SynGR naturally supports zero-shot generalization.
>
> (1) Since SynGR is built on the **same training framework** as MQL4Rec, it inherits its zero-shot transferability to unseen items.
>
> (2) More importantly, by reducing reliance on unimodal shortcuts and encouraging deep cross-modal synergy through Saliency-Aware Masking, SynGR learns more robust multimodal alignments, which **improves generalization to unseen data**.
>
> (3) We further validated this through a zero-shot evaluation on Instruments. As shown below, SynGR consistently outperforms MQL4Rec, suggesting that deeper synergy extraction improves generalization to unseen items.
>
> | Method   | H@10  | N@10  |
> |----------|-------|-------|
> | MQL4Rec  | 0.00099  | 0.00046 |
> | SynGR    | **0.00115**| **0.00061** |
>
> >**Q8: Additional backbones.**
>
>  We further evaluated MACRec and SynGR on two backbones, BART-base (an encoder-decoder model) and Qwen-2.5-0.5B (a decoder-only LLM), using the Instruments dataset.
>
> | Backbone  | Method | H@10 | N@10 |
> | :--- | :--- | :---: | :---: |
> | BART | MACRec| 0.1526 | 0.1051 |
> | BART | SynGR | **0.1645** | **0.1120** |
> | | | | |
> | Qwen-2.5-0.5B | MACRec  | 0.1215 | 0.0756 |
> | Qwen-2.5-0.5B | SynGR | **0.1524** | **0.1042** |

---

> > ### Author Rebuttal · Reviewer_nPWh · 2026-04-03
> >
> > Thank you for the detailed rebuttal.
> > 1 More experiment on different settings and backbone are provided.
> > 2 Generally, multimodal method improve long-tail items more, so these methods achieve better results at metrics like H@50, H@100, but authors only provide experimental results on head-focus metrics.
> >
> >
> > Given these clarifications, I consider my concerns partially resolved and maintain my original score.

---

> > > ### Author Response · Authors · 2026-04-03
> > >
> > > Thank you for your continued feedback！
> > >
> > > (1) First, we sincerely apologize for not including these metrics earlier. Due to the **strict 5,000-character limit** in the previous rebuttal phase, **the 6 newly added experimental tables** (addressing different settings and backbones) almost completely exhausted our space. We had to make a difficult trade-off and unfortunately had **no room left** to report these deeper ranking metrics. We kindly ask for your understanding.
> > >
> > > (2) Second, to directly address your concern, we have evaluated our model and the baselines on **HR and NDCG at @50, @100, and @200**. The detailed results are presented below:
> > >
> > >
> > > ---Table 1----
> > > | Instruments   | H@50  | H@100 | H@200 | N@50 | N@100 | N@200 |
> > > |------------|---------|---------|---------|---------|----------|----------|
> > > | GRU4Rec    | 0.1923  | 0.2515  | 0.3523  | 0.0951  | 0.1021   | 0.1209   |
> > > | BERT4Rec   | 0.1956  | 0.2557  | 0.3585  | 0.0918  | 0.1032   | 0.1211   |
> > > | SASRec     | 0.2089  | 0.2586  | 0.3591  | 0.0989  | 0.1176   | 0.1212   |
> > > | TIGER      | 0.1881  | 0.2512  | 0.3512  | 0.0925  | 0.1002   | 0.1181   |
> > > | MQL4GRec   | 0.2156  | 0.2824  | 0.3611  | 0.1131  | 0.1297   | 0.1455   |
> > > | MACRec     | 0.2213  | 0.2891  | 0.3722  | 0.1163  | 0.1321   | 0.1495   |
> > > | **SynGR**  | **0.2390** | **0.2925** | **0.3974** | **0.1263** | **0.1382** | **0.1500** |
> > >
> > > ---Table 2----
> > >
> > > | Arts   | H@50  | H@100 | H@200 | N@50 | N@100 | N@200 |
> > > |------------|--------|--------|--------|---------|----------|----------|
> > > | GRU4Rec    | 0.2093 | 0.2722 | 0.3322 | 0.0871  | 0.0986   | 0.1033   |
> > > | BERT4Rec   | 0.2121 | 0.2756 | 0.3466 | 0.0886  | 0.0943   | 0.1071   |
> > > | SASRec     | 0.2264 | 0.2789 | 0.3555 | 0.0862  | 0.0993   | 0.1121   |
> > > | TIGER      | 0.2054 | 0.2671 | 0.3301 | 0.0788  | 0.0832   | 0.0956   |
> > > | MQL4GRec   | 0.2204 | 0.2871 | 0.3589 | 0.1020   | 0.1083   | 0.1255   |
> > > | MACRec     | 0.2216 | 0.2855 | 0.3617 | 0.1022  | 0.1093   | 0.1264   |
> > > | **SynGR**  | **0.2345** | **0.3009** | **0.3818** | **0.1169** | **0.1174** | **0.1288** |
> > >
> > > As the results demonstrate, SynGR achieves consistent improvements across all deeper metrics. This empirical evidence directly supports our observation: by unleashing cross-modal synergistic information, our method effectively captures richer item semantics, which is highly beneficial for improving the recommendation quality of long-tail items. We will incorporate these newly added experimental results into the revised version.
> > >
> > > We hope these comprehensive results fully address your remaining concerns.
> > >
> > > Thank you once again for your valuable guidance and time!

---

### Official Review · Reviewer_UU2v · 2026-03-14

**Soundness:** 2
**Presentation:** 3
**Significance:** 2
**Originality:** 2
**Overall Recommendation:** 4
**Confidence:** 4

**Summary:**

This paper targets the research problem of multimodal generative recommendation. To have better cross-modal synergy, it proposes SynGR, a synergistic generative recommendation framework that actively disrupts the path of least resistance during training. Experiments show the effectiveness of this method.

**Compliance With Llm Reviewing Policy:**

Affirmed.

**Final Justification:**

My concerns have been addressed, therefore I consider increase my overal evaluation

**Key Questions For Authors:**

please refer to the weaknesses discussed above

**Limitations:**

There is societal impact statement in this paper, while it lacks the discussion of limitations.

**Strengths And Weaknesses:**

Strengths:
1. Significance: The problem of generative multimodal recommendation has significant research value, especially in the industrial application scenarios.
2. Presentation: The overall presentation is clear and easy to follow, and the writing is well structured and in good shape.

Weaknesses:
1. Originality:
    1.1 The specific research problem of multimodal data fusion is a long-standing research problem, which has also been studied by many papers.
    1.2 The underlying mechanism of multimodal data for recommendation may not be consistent across different domains. In some domains, images are more helpful, in some other domains, text is more important, and in some domains, audio would matter. Therefore, it is skeptical to affirm that less utilization of a certain modality is bad (even though the attention score shown in Figure 1c demonstrates visual is less utilized, this does not mean less utilization of visual data is bad).
    1.3. More details of the PID algorithm used in Figure 1b should be presented. Moreover, the PID framework that assumes the more synergy would yield better performance should also be well justified, is this a well-accepted criteria or conclusion across the community? And if it is a valid and generalizable indicator, it is highly recommended to report this across more baseline methods.
2. Soundness:
    2.1 For baselines of multimodal sequential recommendation, only one work has been compared (MISSRec, TOMM 2023), more recent and strong methods should be compared.
    2.2 For the experiments, can you show some quantitative results and qualitative case studies, to call back and support your motivations (especially Figure 1a) in the Introduction.

---

> ### Author Rebuttal · Authors · 2026-03-30
>
> Thanks a lot for your efforts in reviewing this paper!
>
> >**Q1: Multimodal data fusion has been extensively studied.**
>
> While well-studied in traditional **discriminative** ranking, multimodal fusion faces a fundamentally new challenge in **Generative** Recommendation (GR): models inevitably collapse into unimodal shortcuts, missing critical synergistic semantics.
>
> Our originality tackles this **GR-specific bottleneck** via an active synergy-oriented intervention, deliberately disrupting these shortcuts to force the extraction of higher-order cross-modal dependencies.
>
> >**Q2: Less utilization of a certain modality is bad.**
>
> Thanks! We agree that intrinsic modality importance varies across domains. We **do not claim** "less utilization of a modality is bad". Rather, **highly imbalanced utilization**, driven by unimodal shortcuts, causes a critical loss of cross-modal synergistic information.
>
> Additionally, SynGR **does not blindly** boost visual weights. It dynamically compares modality saliency to suppress whichever modality dominates. For example, in visually-intensive domains like Arts (**Table 1**), vision acts as the shortcut and is masked more frequently (0.5248).
>
> ---Table 1---
>
> | Modality | Instruments | Arts    | Games |
> |----------|-------------|---------|-------|
> | Text     | 0.5943      | 0.4752  | 0.6075 |
> | visual    | 0.4057      | **0.5248** | 0.3925 |
>
> >**Q3: More PID details.**
>
>  (1) The calculation details are as follows:
>
> ---
>
> **Input:**
>  Metric evaluation function $M(\cdot)$
>  Ground truth targets $\mathbf{Y}_{\text{true}}$
>  Predictions: Text-only $\mathbf{Y}_t$, Vision-only $\mathbf{Y}_v$, Joint bi-modal $\mathbf{Y}_j$
>
> **Output:**
>  Normalized PID components: $\mathbf{S}$,  $\mathbf{R}$, $\mathbf{U}_t$, $\mathbf{U}_v$
>
> $P_t \gets M(\mathbf{Y}\_t, \mathbf{Y}\_{\text{true}})$
> $P_v \gets M(\mathbf{Y}\_v, \mathbf{Y}\_{\text{true}})$
> $P_j \gets M(\mathbf{Y}\_j, \mathbf{Y}\_{\text{true}})$
>
> $\mathbf{S} \gets \dfrac{\max(0,\; P_j - \max(P_t, P_v))}{P_j}$
>
> $\mathbf{R} \gets \dfrac{\min(P_t, P_v)}{P_j}$
>
> $\mathbf{U}_t \gets \dfrac{P_t - \min(P_t, P_v)}{P_j}$
>
> $\mathbf{U}_v \gets \dfrac{P_v - \min(P_t, P_v)}{P_j}$
>
> **Return** $\mathbf{S}, \mathbf{R}, \mathbf{U}_t, \mathbf{U}_v$
>
> ---
>
> (2) SynGR **does not** operate on the premise that more synergy **always** yields better results. Instead, our core argument is that in current GR models, the synergistic component is **under-utilized** and overshadowed by unimodal shortcuts.
>
> This perspective is widely accepted [1,2]: redundancy ensures robustness, but synergy unlocks the true potential of multimodal integration by capturing emergent properties that neither modality possesses alone. In recommendation scenarios, complex item semantics, such as aesthetic appeal, require deep cross-modal reasoning.
>
> (3) As suggested, we verified the PID distribution in more models:
>
> ---Table 2---
>
> | Method |  $\mathbf{S}$ | $\mathbf{R}$ |  $\mathbf{U}_t$ |  $\mathbf{U}_v$ |
> | :--- | :---: | :---: | :---: | :---: |
> | MQL4Rec | 6.4% | 63.5% | 20.2% | 9.9% |
> | MACRec | 7.5% | 60.0% | 20.0% | 12.5% |
> | SynGR | **29.1%** | 45.0% | 16.0% | 9.9% |
>
> **Reference**:
>
> [1] InfMasking: Unleashing Synergistic Information by Contrastive Multimodal Interactions, NeurIPS'25
>
> [2] What to Align in Multimodal Contrastive Learning? ICLR'25
>
> >**Q4: More baselines.**
>
>  We **conducted additional experiments** on Games comparing SynGR with two recent SOTA baselines: FindRec and CAMMSR.
>
> As shown in **Table 3**, SynGR also achieves improvements over these two baselines, further validating its effectiveness.
>
>  We will include these updated baseline comparisons in final version.
>
> ---Table 3---
>
> | Model | H@5 | H@10 | N@5 | N@10 |
> |-------|------|-------|--------|---------|
> | FindRec (KDD'25) | 0.0669 | 0.0980 | 0.0397 | 0.0412 |
> | CAMMSR (ICDE'26) | 0.0674 | 0.1033 | 0.0422 | 0.0501 |
> | SynGR | **0.0702** | **0.1092** | **0.0471** | **0.0596** |
>
> >**Q5: Quantitative and qualitative results for Figure 1a.**
>
> (**Quantitative**) To quantitatively this, we evaluated the PID distribution across three models (**Table 2**). The results confirm that baselines lack synergy and default to textual redundancy, corroborating the shortcut phenomenon highlighted in Figure 1(a).
>
> (**Qualitative**) Please refer to [case study.](https://anonymous.4open.science/r/syngr-633C/case.pdf)
>
> >**Q6: Lacks the discussion of limitations.**
>
> Due to rebuttal space limits, we briefly summarize two main limitations here, which will be fully detailed in the final manuscript.
>
> (1) While SynGR effectively captures item-side cross-modal synergy, it does not explicitly incorporate user-item collaborative filtering signals. Integrating such interaction-driven data may further enhance recommendation accuracy.
>
> (2) This study primarily validates the framework using textual and visual information. Future work could explore its generalizability across a broader range of modalities, such as audio and video.

---

> > ### Author Rebuttal · Reviewer_UU2v · 2026-04-05
> >
> > Thanks for the rebuttal and I increase my overall evaluation score.

---

> > > ### Author Response · Authors · 2026-04-05
> > >
> > > We sincerely appreciate your insightful reviews and positive feedback!

---

### Decision · Program_Chairs · 2026-04-30

**Decision:**

Accept (regular)

**Comment:**

After carefully considering the four reviews, I recommend a “weak accept” for this paper. The reviews collectively acknowledge the paper’s strengths, including its novel focus on synergistic information beyond simple modal alignment, a solid theoretical grounding using Partial Information Decomposition (PID), and clear empirical gains on benchmark datasets. However, several consistent concerns temper this enthusiasm. The reviewers note that the experimental evaluation is limited to Amazon product datasets, leaving generalizability to other domains or larger-scale data unproven. The core technical contribution—the saliency-aware masking strategy—is viewed by some as incremental. Furthermore, questions remain regarding the handling of conflicting modality signals and the method’s scalability beyond text-image pairs. While the paper is technically solid and advances a relevant sub-area, these limitations in evaluation breadth and the somewhat narrow scope of the contribution prevent a stronger recommendation.